# Indonesia Tornado Database: Tornado Climatology of Indonesia

Irfans Maulana Firdaus<sup>1</sup>, Takeshi Yamazaki<sup>1</sup>, Muhammad Rais Abdillah<sup>2</sup>, and Edi Riawan<sup>2</sup>

**Correspondence:** Irfans Maulana Firdaus (firdaus.irfans.maulana.q6@dc.tohoku.ac.jp)

**Abstract.** The climatology of tornadoes, waterspouts, and funnel clouds for Indonesia is compiled into a new tornado database based on newspaper archives, online news, and social media (X, YouTube), covering the period from 1800 to 2024. We present the analysis over two periods: (i) historical and (ii) recent periods. The climatology includes the annual, monthly, diurnal, and geographical data for tornado cases. Based on a review of 47,669 reports mentioning tornadoes, we identified 436 tornado cases with sufficient evidence to be classified as tornado events. In the recent period (2010–2024), the annual frequency of tornadoes was 16.20 cases/year, while in the historical period (1800–2009), the annual frequency was 1.10 cases/year. Tornadoes were mostly documented in Java Island, followed by Sumatra and Sulawesi. The monthly variability of tornadoes shows a maximum during November, followed by December and January. The peak of the diurnal cycle of tornado cases is between 1300 and 1700 Local Solar Time.

#### 10 1 Introduction

Tornadoes are among the most destructive meteorological phenomena, causing significant damage to buildings, infrastructure, and the surrounding landscape. The climatology of tornadoes is crucial for understanding the dynamics and characteristics of tornado occurrences, as well as for enhancing risk assessment and mitigation strategies. Knowledge of tornado characteristics can enhance community response to the tornado threats and help identify vulnerable populations (Johnson et al., 2021). Several open databases have been published online by countries, including the European countries (European Severe Weather Database or ESWD, Dotzek et al. (2009); https://eswd.eu/), the United States (National Centers for Environmental Information, NCEI; www.ncdc.noaa.gov/), Canada (Northern Tornadoes Project, NTP; https://uwo.ca/ntp/), and Japan (Japan Meteorological Agency, JMA; https://www.data.jma.go.jp/stats/data/bosai/tornado/list.html).

On the other hand, tornado climatologies have been published for several countries across Europe (Antonescu et al., 2016), Northern Eurasia (Chernokulsky et al., 2020), South America (Veloso-Aguila et al., 2024), Australia (Kounkou et al., 2009), and Asia countries such as Japan (Niino et al., 1997; Kawazoe et al., 2023), China (Chen et al., 2018; Zhang et al., 2023), India-Pakistan (Bhan et al., 2016), and the Philippines (Capuli, 2024). Antonescu et al. (2016) summarizes the tornado observation for 30 countries across Europe (see their Appendix A). In addition to local meteorological agency sources for tornado reports, other primary sources include old newspaper archives, online news, and social media (Newark, 1984; Paul, 2001; Rauhala et al., 2012; Kahraman and Markowski, 2014; Matsangouras et al., 2014; Antonescu and Bell, 2015; Veloso-Aguila et al., 2024). The

<sup>&</sup>lt;sup>1</sup>Graduate School of Science, Tohoku University, Sendai, Japan

<sup>&</sup>lt;sup>2</sup>Atmospheric Sciences Research Group, Faculty of Earth Sciences and Technology, Institut Teknologi Bandung, Bandung, Indonesia

**Figure 1.** (a) Tornado in Rancaekek, West Java, on 21 February 2024 (Jawa Pos, 2024), (b,c) damage caused by a tornado in Rancaekek (Tempo, 2024; RRI, 2024), (d) Tornado in Bogor City, West Java on 6 December 2018 (Hidayatullah, 2018), (e,f) damage caused by a tornado in Bogor City (Tempo, 2018; Kompas, 2018). The figures are downloaded from news article online.

term for a tornado may vary from one country to another. For example, words such as *kasırga*, *firtına*, and *siklon* are described as tornado events in Turkey (Kahraman and Markowski, 2014). Moreover, in Romania, they use words like *uragan/orcan*, *trombă*, *tornadă*, and *vârtej* to describe a tornado events.

The occurrence of tornadoes in Indonesia is rare (Maas et al., 2024) compared to mid-latitudes regions such as the United States. Nevertheless, their impact can still be devastating. Figure 1 shows the tornado cases that occurred in Indonesia. A tornado hit Rancaekek, Sumedang, West Java on 21 February 2024, damaged over 1,000 buildings and caused at least 47 injuries (Figure 1a-c). On 6 December 2018, a tornado occurred in Bogor, West Java, resulting in damage to 854 buildings (Figure 1d-f). However, tornado reports in Indonesia have not been systematically recorded, unlike in countries such as the United States and those in Europe.

35

This paper aims to identify tornado cases in Indonesia from numerous scattered reports and to investigate basic climatological characteristics of the tornadoes in Indonesia. The identified tornadoes are published in an open repository that can catalyze future tornado studies in the tropics. This paper is organized as follows: Section 2 presents the data and methodology related to the collection and categorization of tornado reports for the database. Section 3 presents the result of the tornado climatology for Indonesia. Section 4 presents a discussion about the limitation, the relationship between convective precursor and tornado, and tornado rating differences. The last section, section 5 provides a conclusion.

Table 1. Credibility and category tornado reports.

| Category  | Criteria                                                                                                                                                                                |  |
|-----------|-----------------------------------------------------------------------------------------------------------------------------------------------------------------------------------------|--|
| Confirmed | 1. A photograph or video of a tornado 2. Damage survey of tornado                                                                                                                       |  |
| Probable  | 1. Eyewitness testimony who reported seeing rotation wind and/or column air directly     2. Eyewitness report of typical tornado damage     3. A photograph of a typical tornado damage |  |
| Possible  | Eyewitness testimony who reported hearing thunder directly     There was hail     Cause of the damage is not confirmed by the observations of an eyewitness                             |  |

#### 2 Data and Methods

55

This section describes the development of the tornado database for Indonesia. The term 'tornado' is not commonly used in Indonesia, as it has a local name, *angin puting beliung*, which translates to 'rotating wind'. In this article, we use the term 'tornado' instead of *angin puting beliung* and address the criteria and definition of a tornado. With a clear definition of tornadoes, we created a new tornado database by collecting historical reports from archives, including newspapers and social media. Finally, we analyze the climatology of tornadoes in Indonesia.

The Meteorological, Climatological, and Geophysical Agency of Indonesia (BMKG RI) defines a tornado as "a rotating, strong wind originating from a cumulonimbus cloud, occurring for a short period". Meanwhile, the National Disaster Management Authority of Indonesia (BNPB RI) defines a tornado as "a sudden, rotating, strong wind with a center and spiral-like appearance, extending to the ground and dissipating within a short time (3-5 minutes)". Given these differing definitions, we address the definition of a tornado. In this article, the tornado definition is adopted from American Meteorological Society (2024). A tornado is defined as "a rapidly rotating column of air that extends from the surface to the base of a cumuliform cloud and is often visible as a funnel cloud". Additionally, this definition is extended to include all waterspouts, whether or not they make landfall, in line with the tornado definition by Rauhala et al. (2012).

The criteria for evaluating tornado cases were adopted and modified from Rauhala et al. (2012) (Table 1). We made additions and modifications to the 'probable' and 'possible' categories. An eyewitness testimony that reported observing rotating winds and/or a column of air directly was included in the 'probable' category, while an eyewitness testimony that reported hearing thunder and observing hail directly was placed in the 'possible' category. A report is categorized as a tornado case if any of the criteria are satisfied. All criteria were accepted in the new database, but only confirmed and probable tornado cases were analyzed for tornado climatology in Indonesia (Rauhala et al., 2012; Kahraman and Markowski, 2014).

The tornado climatology in Indonesia was divided into two different periods. The first period is the historical period, starting in 1800 and ending in 2009. The second period is the recent period, spanning from 2010 to 2024, following the release of the

**Figure 2.** Period of tornado reports sources. There are four main sources: (i) the Dutch old newspaper archives, (ii) National Library and Press Museum, (iii) BMKG RI extreme weather database, and (iv) Social media.

BMKG regulation on the "Standard Operating Procedure for Early Warning Implementation, Reporting, and Dissemination of Extreme Weather Information", as well as the period during which the Internet became widely used. The periods of tornado report sources are illustrated in Figure 2.

## 2.1 Historical period

65

Tornado reports for the historical period were collected from old newspaper archives. There are two sources used: (1) Dutch old newspaper archives (1800-1957), which were accessed via the online website (https://www.delpher.nl/) and (2) the National Library and Press Museum newspaper archives (1945-2009), which were accessed via the online website (https://mpn.kominfo.go.id/arsip/; http://khastara.perpusnas.go.id). The Dutch newspaper archives were searched using three keywords: (a) tornado, (b) windhoos, and (c) wervelwind. We filtered the archive newspapers by region, specifically *Nederlands-Indie/Indonesie*. The archived newspapers included *Java Govt. Gazette, De Locomotief, Javasche Courant, Deli Courant, Soerabaiasch-Handelsblad, Batavia Nieuwsblad, De Preangerbode, De Indische Courant, Het Nieuws Van Den Dag, De Sumatra Post, Java Bode, Sumatra-Courant, etc.* The newspaper archives from the National Library and Press Museum were searched using two keywords: (a) *angin puyuh* and (b) *angin puting beliung*.

In total, 9,930 reports were collected from the Dutch old newspaper archives and 18,958 reports from the National Library and Press Museum newspaper archives (Table 2). A total of 193 tornado reports were accepted into the database. The first recorded tornado case in Indonesia was reported by the *Java Govt. Gazette*. The tornado occurred in Surabaya, East Java, on 7 December 1815. The tornado passed through the town, resulting in one house being destroyed, with one fatality and four people injured. However, this case was categorized as a possible tornado. The first probable tornado case occurred in Temanggung, Central Java, on 30 December 1834. The tornado struck a large tree, destroyed 70 houses, and damaged rice fields. Fortunately, there were no fatalities in this disaster.

Table 2. Total tornado reports and total tornado cases were identified.

| No. | Source                            | Total reports | Total tornado cases |
|-----|-----------------------------------|---------------|---------------------|
| 1   | Dutch old newspaper archives      | 9,330         | 173                 |
| 2   | National Library and Press Museum | 18,958        | 36                  |
| 3   | BMKG RI                           | 1,832         | 70                  |
| 4   | Social Media                      | 17,549        | 157                 |
|     | Total                             | 47,669        | 436                 |

## 2.2 Recent period

The primary sources for the recent period were the newspaper archives of the National Library and Press Museum (2010-2024), the BMKG RI extreme weather database (2012-2024), and social media (2010-2024, including X, YouTube, and News). The BMKG RI extreme weather database collects internet news and compiles it into an online database, accessible at https://pikacu.bmkg.go.id/. Tornado reports from social media platforms, such as X, were used to improve the quality of the reports (Baranowski et al., 2020). The tornado cases were extracted through data mining using Python and Generative AI (Gemini Application Programming Interface - API) to eliminate noisy data, such as metaphorical terms or non-weather-related usage. The keywords used for mining the tornado cases were similar to those used in the old newspaper archives. For each identified case, the corresponding social media URLs were collected to enable manual verification and validation. Any unrelated or unverifiable sources were excluded from the dataset. After that, we categorized the tornado cases based on the criteria in Table 1. In cases where multiple sources reported the same event, the report was classified based on the most reliable source, preferably those accompanied by photographs, videos, or damage surveys. In total, there were 1,832 reports from the BMKG RI extreme weather database and 17,549 reports from social media (Table 2). During the recent period, 243 tornado cases were added to the database.

#### 2.3 Indonesia Tornado Database

Indonesia has no tornado database yet. While the BMKG RI provides an extreme weather database via its online website at https://pikacu.bmkg.go.id/, and the BNPB RI maintains a disaster database that can be accessed online at https://gis.bnpb.go.id/, neither offers detailed information on tornado cases. The BMKG RI's extreme weather database collects news from the internet and compiles it with other disaster data, such as floods, landslides, hail, strong winds, and heavy rainfall. On the other hand, the BNPB RI disaster database does not explicitly mention tornadoes, as tornadoes are categorized under extreme weather events, alongside strong winds, hail, tropical cyclones, and extreme temperature changes. Therefore, we developed the Indonesia Tornado Database to document the tornado cases that have occurred in Indonesia (Firdaus and Iswahyudi, 2025). Table 3 shows the attributes of the Indonesia Tornado Database.

Table 3. Structure of Indonesia Tornado Database

| Field name                | Description                                           | Field Type                               |
|---------------------------|-------------------------------------------------------|------------------------------------------|
| ID                        | Identifier of a tornado                               | Number                                   |
| Туре                      | Type of a tornado                                     | List: F for Funnel Cloud, T for Tornado, |
|                           |                                                       | WS for Waterspout                        |
| Category                  | Categorization of a tornado                           | List: Confirmed, Probable, Possible      |
| Date                      | Date of a tornado                                     | Date                                     |
| Time (LST)                | Time of a tornado                                     | Time in Local Solar Time (LST)           |
| Time (UTC+7)              | Time of a tornado                                     | Time (UTC+7)                             |
| Province                  | Province of a tornado occurred                        | String                                   |
| Regency                   | Regency of a tornado occurred                         | String                                   |
| District                  | District of a tornado occurred                        | String                                   |
| Village                   | Village of a tornado occurred                         | String                                   |
| Lon                       | Longitude                                             | Number                                   |
| Lat                       | Latitude                                              | Number                                   |
| Building Damage           | Dividing into three categories: Severe, Moderate,     | Number                                   |
|                           | Slight. The total number of building damage caused by |                                          |
|                           | a tornado                                             |                                          |
| Injuries                  | Dividing into two categories: Death and Injuries. The | Number                                   |
|                           | total number of injuries caused by a tornado          |                                          |
| Economic Loss (Approxima- | The approximation of economic loss caused by a tor-   | Number (in Indonesia currency: Ru-       |
| tion)                     | nado                                                  | piah)                                    |
| Source                    | The source of tornado reports                         | List: Website, National Library & Press  |
|                           |                                                       | Museum, BMKG RI, and Social Media        |
| Website                   | The source website                                    | String                                   |

## 3 Results

# 3.1 Decadal and Annual Frequency

The total number of tornado cases in Indonesia, including confirmed and probable cases, from 1800 to 2024, is about 436 cases. More than half of these cases (243 cases) occurred in the last 15 years (Fig. 3). The increasing trend in tornado cases is likely attributed to advances in technology, such as the widespread use of the internet and social media. Therefore, it is not possible to determine whether tornadoes, as a natural hazard, are increasing in frequency.

In the historical period, the total number of tornado cases is 193, which occurred between 1834 and 2009 (175 years). It means the average number of tornado cases is 1.10 tornadoes per year. There was a significant increase in tornado reports from

**Figure 3.** Annual frequency of tornado cases and number of tornado reports: (a) Historical period from 1830s until 2000s (per decades) and (b) Recent period from 2010 until 2024.

the 1830s to the 1870s, followed by another rise from the 1880s to the 1950s. This increase is presumed to be due to the growing number of newspaper publishers during that period (from 15 publishers to 20 publishers). A break in tornado cases is found from the 1960s to the 1970s. We speculate that this gap resulted from a shift from Dutch newspapers to Indonesian newspapers, which led to changes in the management of newspaper content. After the 1980s, tornado reports remained relatively constant until the 2000s.

On the other hand, the total number of tornado cases is about 243 tornadoes between 2010 and 2024 (15 years), showing an increase during this period. The average number of tornadoes is about 16.20 tornadoes per year. A notable difference is observed between the periods before and after 2016. We speculate that the increase in tornado cases is due to the widespread use of the internet and social media. Additionally, public awareness of extreme weather events has increased, which has likely influenced the tornado database in recent years (Antonescu et al., 2016).

**Figure 4.** Percentage spatial distribution of tornado cases in Indonesia: (a) Historical period from 1830s until 2000s (per decades) and (b) Recent period from 2010 until 2024.

## 3.2 Spatial Distribution

The spatial distribution of tornado cases is shown in Figure 4 for six regions: (i) Sumatra, (ii) Java, (iii) Borneo, (iv) Sulawesi, (v) Bali-Nusa Tenggara, and (vi) Maluku-Papua. In our database, tornado cases are most frequent in Java for both periods. The lowest occurrence of tornadoes was reported in the Maluku-Papua region. According to historical reports, over 75% of tornado reports originated from Java, with fewer reports from northern Sumatra, eastern Borneo, central Sulawesi, and the Bali-Nusa

Tenggara region. No tornado reports were recorded from Maluku-Papua. We speculate that the higher frequency of tornado reports in Java and the lower frequency in certain regions are due to: (i) the newspaper publishers and (ii) the population density. The old newspaper archives were primarily found in large cities, such as *Java Govt. Gazette* from Batavia (now Jakarta) in Java, *De Locomotief* from Surabaya in Java, *Deli Courant* from Surabaya in Java, *De Preangerbode* from Bandung in Java, *De Sumatra Post* from Medan in Sumatra, and *Sumatra-Courant* from Padang in Sumatra. Additionally, Indonesia's population is not evenly distributed (Figure S1). Over 63.8% of Indonesia's population lived in Java, and 17.4% lived in Sumatra.

In the recent period, tornado reports have been more evenly distributed compared to the historical period, although tornado reports remain most frequent in Java. There has been an increase in tornado reports across Sumatra, Borneo, Sulawesi, and Bali-Nusa Tenggara. The usage of the internet and population growth are the main reasons for the increase in tornado reports during this period (Anderson et al., 2007). The population growth in Sumatra, Borneo, Sulawesi, and Maluku-Papua has increased by 2-5 times between 1971 and 2024 (Figure S2). Over the past 15 years, the internet and social media have been widely used globally, including in Indonesia. For example, the number of X users in Indonesia was about 29.4 million in 2012 (Carley et al., 2015), and this number is expected to have increased since then. This widespread use of the internet means that information, including disaster reports, can be disseminated easily (Baranowski et al., 2020).

# 3.3 Monthly and Diurnal Distribution

135

Figure 5 shows the monthly and diurnal distribution of tornado cases in Indonesia from 1834 to 2024. Approximately 64% of tornado cases occurred between November and March (Figure 5a). This pattern is coinciding with the Asia-Australia monsoon system, which is active during the boreal winter (Wheeler and McBride, 2005). Tornadoes occurred most frequently in November and least frequently in June.

The percentage of monthly distribution for each region is shown in Figure 5b. The monthly peak of tornado cases varies by region. The peak differences among regions coincide with the precipitation pattern in Indonesia (Aldrian and Susanto, 2003). Java and Bali-Nusa Tenggara have a peak of tornado cases during the Nov-Dec-Jan-Feb season, while Maluku-Papua has a peak between May-Jul, and Borneo has a peak between Jul-Aug. Sulawesi has peak tornado cases in March, September, and December. Sumatra has peak tornado cases in April, August, and November. These patterns are a result of the distinct precipitation patterns characteristic of these islands.

The diurnal distribution of tornado cases in Indonesia was placed into 1-hour bins in Local Solar Time (LST): 0000-0059, 0100-0159, etc (Figure 5c). The old newspaper archives sometimes do not provide detailed information regarding the timing of tornado occurrences. Therefore, we categorized that data into five terms: Mor (morning), Aft (afternoon), Eve (evening), Nig (night), and NaN (no information available). Tornadoes were most frequently reported during the daytime between 1300-1700 LT and in the evening, with approximately 56% of tornado cases occurring during this time. A smaller portion of tornado cases occurred in the early morning and at night. This pattern coincides with the diurnal cycle of precipitation in Indonesia. Peatman et al. (2021) stated that the diurnal cycle of precipitation in Indonesia is driven by temperature contrasts between land and sea, which create deep convection in the afternoon. Additionally, the complex topography in Indonesia can influence and enhance convection through orography (Peatman et al., 2021; Firdaus et al., 2024).

**Figure 5.** (a) Monthly distribution, (b) percentage of monthly distribution based on different regions, and (c) diurnal distribution of tornado cases in Indonesia for the historical and recent period. The percentage is calculated by dividing the number of tornado cases for a specific month and region by the total number of tornado cases in that region. In the inset of panel (c), 'Mor', 'Aft', 'Eve', 'Nig', 'NaN' represent morning, afternoon, evening, night, and missing data, respectively.

# 4 Discussion and Conclusion

#### 4.1 Limitation

The tornado cases were collected from four primary sources: (i) old newspaper archives from the website https://delpher.nl/, (ii) the National Library and Press Museum, (iii) the BMKG RI extreme weather database, and (iv) Social media. We realized that these sources can lead to a bias in the spatiotemporal distribution of tornado cases in Indonesia. Not all the old newspapers, both from the website and the National Library and Press Museum, were available for analysis. The description of the event was also limited, which could result in errors regarding location and time. The coverage of the old newspaper could result in bias in the spatial distribution of tornado cases, as we speculate in Section 3.2 that the old newspapers were primarily found in the big cities, such as Padang and Medan in North Sumatra, and Jakarta, Surabaya, and Bandung in Java. Reports from old newspapers may also have been influenced by editorial choices or public interest. The BMKG RI extreme weather database and social media could also result in bias, especially in spatial events, because it depends on the observer or people who reported the tornado. The population density in Indonesia is not distributed spatially (see Supplementary Files, Figure 1). For example, the Maluku-Papua and the central Borneo have population density around 0-50 person/km² compared to other regions, such as Java, that have a population density of over 500 person/km².

#### 4.2 Influence of Convective Precursors

Previous studies have examined the influence of convective precursors on tornadic storms, such as the Madden-Julian Oscillation (MJO) (Barrett and Gensini, 2013; Tippett, 2018; Veloso-Aguila et al., 2024) and the El Niño-Southern Oscillation (ENSO) (Allen et al., 2015). Maritime Continent (MC) convective activity, especially Indonesia, is influenced by various phenomena, such as MJO, ENSO, Indian Ocean Dipole (IOD), monsoon, and cold surge (Yoneyama and Zhang, 2020). Muhammad et al. (2021) showed that during phases 2 through 4 of active MJO, convection develops in the MC and can increase the extreme precipitation probability. Kurniadi et al. (2021) found that ENSO and IOD can affect extreme precipitation in Indonesia. The impact of ENSO on extreme precipitation is more pronounced between June and November, but less significant from December to May. In contrast, the IOD has a significant effect on extreme precipitation only during the June to November period. Chang et al. (2005) showed that monsoon can influence the increasing and decreasing precipitation in MC. Cold surges also influence the precipitation in MC (Hattori et al., 2011). These phenomena modulate convective activity, which can subsequently lead to extreme precipitation. However, there are limited studies that specifically examine how synoptic-scale conditions modulate tornado activity in tropical regions. Therefore, further investigation is needed to assess the influence of convective precursors on tornado occurrence in Indonesia.

## 4.3 Indonesian Tornado Rating

The rating of tornadoes is critically important for understanding and mitigating the impacts of these severe weather events. Tornado ratings typically use the Fujita scale (Fujita, 1973) or the T-Scale (Meaden, 1976; Feuerstein et al., 2011), both of

which are based on the severity of damage caused by strong winds. The Fujita Scale was later updated to the Enhanced Fujita Scale (EF-Scale) in 2007 (McDonald and Mehta, 2006). The EF-Scale utilizes damage indicators (DIs), with each DI having several degrees of damage (DODs). Various adaptations of the EF-Scale have been developed in different countries, such as in Canada (Sills et al., 2014; Environment Canada's Weather Service, 2018), known as the EF-Scale Canada, and in Japan (Japan Meteorological Agency, 2015), referred to as the Japanese EF-Scale (JEF-Scale). These adaptations modify the DI and DOD based on regional factors such as building structures, vegetation, and construction materials, which vary by country.

In Indonesia, there is currently no standardized procedure to assess tornado damage, and as a result, tornadoes in the country are not rated. Two recent tornado events, the Tornado Bogor (2018) and Tornado Rancaekek (2024) (Figure 1), did not have a rating. Figure 1 shows the appearance of the Tornado Rancaekek and Tornado Bogor with the damaged buildings and trees caused by these events. The Tornado Rancaekek damaged 1,177 houses and buildings, while the Tornado Bogor damaged 1,697 houses and buildings.

Table 4 presents the tornado assessment for Tornado Rancaekek and Tornado Bogor based on both the EF-Scale and the JEF-Scale. Both scales use the DI and DOD to assess the expected wind speed of a tornado, assigning ratings based on the highest wind speed. According to the EF-Scale, Tornado Rancaekek can be categorized as EF2, as it caused significant damage to a factory building, while Tornado Bogor is categorized as EF1, due to the loss of roof covering materials on buildings. On the other hand, according to the JEF-Scale, Tornado Rancaekek is classified as JEF3, while Tornado Bogor is categorized as 210 JEF1.

It is clear from the evidence above that there is a difference between the EF-Scale and the JEF-Scale, which is caused by their DI and DOD classifications. Indonesia has distinct building and structural characteristics compared to countries such as the United States and Japan. For instance, many houses in Indonesia are constructed with wood and brick, and roofs are often made of tiles. These factors influence the DOD classification, which can lead to overestimation of the tornado's rating when applying the EF-Scale or JEF-Scale. Therefore, it is recommended that Indonesia develop its tornado scale, tailored to its unique construction practices, to assess and rate tornado events in Indonesia more accurately.

# 5 Conclusion

This paper presents what is believed to be the first climatological analysis of tornado cases in Indonesia to date. The tornado cases were collected from four primary sources from 1800 to 2024: (i) old newspaper archives from the website https://delpher.nl/, (ii) the National Library and Press Museum, (iii) the BMKG RI extreme weather database, and (iv) Social media. The tornado climatology was divided into two periods: (i) historical period (1800-2009) and (ii) recent period (2010-2024). The main conclusion of the study can be briefly summarized as follows:

A total of 436 tornado cases occurred from 1800 to 2024. In the historical period (1800-2009), tornadoes were reported in 193 cases, or approximately one tornado/year. In the recent period (2010-2024), there were 243 cases, or 16-17 tornadoes/year.

Table 4. Tornado Rancaekek and Bogor assessment (Figure 1).

| No. | Tornado   |                                                                                                                                                                                                                           | Assessment                                                                                                                                                                                                                 |
|-----|-----------|---------------------------------------------------------------------------------------------------------------------------------------------------------------------------------------------------------------------------|----------------------------------------------------------------------------------------------------------------------------------------------------------------------------------------------------------------------------|
|     |           | EF-Scale                                                                                                                                                                                                                  | JEF-Scale                                                                                                                                                                                                                  |
| 1.  | Rancaekek | (Figure 1b) DI 23: Warehouse Building DOD 5: Collapse of other non-bearing exterior walls Expected wind speed: 114 miles/hour  (Figure 1c) DI 28: Trees-Softwood DOD 4: Trunk snapped Expected wind speed: 104 miles/hour | DI 2: Industrialized steel-framed houses (prefabricated) DOD 5: Deformation/loss of wall cladding Expected wind speed: 75 m/s  DI 25: Broad-leaved trees DOD 3: Trunk snapping (without decay) Expected wind speed: 60 m/s |
|     |           | EF-Scale: EF-2                                                                                                                                                                                                            | JEF-Scale: JEF-3                                                                                                                                                                                                           |
|     |           | (Figure 1e) DI 1: Small Barns and Farm Outbuildings DOD 4: Major loss of roof panels Expected wind speed: 90 miles/hour                                                                                                   | DI 1: Wooden houses and stores DOD 4: Destruction/detachment of eaves or sheathing roof boards Expected wind speed: 50 m/s                                                                                                 |
| 2.  | Bogor     | (Figure 1f) DI 28: Trees-Softwood DOD 3: Trees uprooted Expected wind speed: 87 miles/hour                                                                                                                                | DI 25: Broad-leaved trees  DOD 2: Uprooting without root decay  Expected wind speed: 45 m/s                                                                                                                                |
|     |           | EF-Scale: EF-1                                                                                                                                                                                                            | JEF-Scale: JEF-1                                                                                                                                                                                                           |

- The most frequent tornado reports were reported in Java for both periods, followed by Sumatra, Sulawesi, Bali-Nusa
   Tenggara, Borneo, and Maluku-Papua. It is speculated due to population density and the widespread use of the internet.
- Tornado cases mainly occurred during Nov-Dec-Jan-Feb-Mar, coinciding with the precipitation pattern in Indonesia.
   The majority of tornado cases occurred between 1300 and 1700 LT and in the evening.
- The Indonesian tornado database developed in this study can help fill the information gap regarding severe weather, particularly tornadoes, and provide a better understanding of tornadoes in the tropical maritime region. The next step in exploring tornadoes in Indonesia is to examine the characteristics of environmental conditions associated with tornadoes, as this is crucial for improving prediction and risk assessment in Indonesia.

Data availability. The Indonesia Tornado Database is available at ttps://doi.org/10.5281/zenodo.15099438 (Firdaus and Iswahyudi, 2025).

The old newspaper archives were collected and saved by the author. These are available upon request from the corresponding author on reasonable request.

Author contributions. IMF: conceptualization, data curation, methodology, writing (original draft). TY: supervision, writing (review and editing). MRA: supervision, writing (review and editing). ER: supervision, writing (review and editing).

Competing interests. The contact author has declared that none of the authors has any competing interests.

240 Acknowledgements. The first author was funded by a scholarship from Japan International Cooperation Agency (JICA) under Human Resources Development in Science, Technology and Innovation Program. The first author would like to thank National Press Museum and National Library who provide the old newspaper archives. Thank to Ridha Fatony Iswahyudi for helping the first author to collect the tornado reports. This study is also partly funded by "Program Riset ITB tahun 2025" (grant number: 533E/IT1.A/SK-KP/2025) and "Program PPMI FITB 2025" (file number: FITB.PPMI-1-01-2025). Finally, we extend thanks to the anonymous reviewers for their invaluable feedback and suggestions.

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
