# Peer review of "Indonesia Tornado Database: Tornado Climatology of Indonesia"

_EGUsphere, 2025_

## Author Comment (AC1)

**Response to Reviewer #1**

We truly express our appreciation for the time and the effort that you dedicated to providing feedback on our work. We are grateful for the insightful comments and valuable suggestions that have significantly improved our work. We have incorporated the comments and recommended suggestions. Please refer to the section below, marked in blue, for a detailed response to the reviewers' comments and concerns, with page numbers corresponding to the revised manuscript.

**General Comment:**

The manuscript presents the first comprehensive tornado climatology for Indonesia, compiling 436 events from 1834–2024 using multiple data sources. The work fills an important gap in our current knowledge, because unlike the US or Europe, Indonesia had no tornado database, so publishing an analysis of confirmed and probable cases is a valuable contribution. The manuscript is in general well-structured and mostly clear in writing. The inclusion of historical archives, Indonesian Met Service records, and social-media/news reports demonstrates effort and to some extent novelty. The climatological findings are potentially of interest for regional hazard assessment. At the same time, the authors should be very cautious in interpreting the rising trend historical vs. recent period as anything other than improved reporting. Overall, the manuscript is a good contribution to tornado climatology in Southeast Asia, but some contextual analysis is lacking (see below)

**Reply:** We thank to reviewer for recognizing our work for providing the Indonesian Tornado Database and providing comments and suggestions. We provide point-by-point responses below.

**Specific Comments**

1. The authors note that modern reports (2010–2024) dominate the database due to Internet/social-media proliferation. This implies strong observational bias (i.e., underreporting before the digital era). The spatial climatology likely reflects population density and media coverage rather than the true distribution of tornadoes. It might help to explicitly state how duplicate reports were merged (e.g., when one

tornado generated multiple news reports). The use of "generative AI" (Gemini) to filter social media reports is useful, but a brief note on validation or potential biases of this method would strengthen confidence in the results. Lastly, ensure all non-English terms are clearly translated when first used.

**Reply:**

We acknowledge that our work is subject to a bias in the spatiotemporal distribution of tornado reports. Not all the old newspapers, both from the website and the National Library and Press Museum, were available for analysis. Thus, it sets the limit of tornado reports in the historical period. The population density also introduces a bias in spatial climatology due to the people who reported the tornado. The population density of Indonesia is not evenly distributed, resulting in bias. However, despite these limitations, the database remains valuable for understanding the temporal and spatial patterns of tornado occurrence in Indonesia. The use of strict classification criteria and multiple data sources helps enhance the reliability of the tornado records compiled in this study.

We add the explanations about how multiple data sources reported the same event in subsection 2.2 line 92 → "*In cases where multiple sources reported the same event, the report was classified based on the most reliable source, preferably those accompanied by photographs, videos, or damage survey*".

We add the brief note about the using of Gemini AI in subsection 2.2 line 87 → "*The tornado cases were extracted through data mining using Python and Generative AI (Gemini Application Programming Interface - API) to eliminate noisy data, such as metaphorical terms or non-weather-related usage. The keywords used for mining the tornado cases were similar to those used in the old newspaper archives. For each identified case, the corresponding social media URLs were collected to enable manual verification and validation. Any unrelated or unverifiable sources were excluded from the dataset.*"

2. The classification scheme (Table 1) follows Rauhala et al. (2012) and similar studies. One suggestion is to ensure the terms like "credible eyewitness" are well-defined. The wording in Table 1 could be tightened (e.g., "credible eyewitness who reported hearing a thunderous sound" should be "reported hearing thunder"). It might be worthwhile to note explicitly that without damage surveys, classification

from media accounts is inherently uncertain, for example, some "probable" tornadoes may have been straight-line wind events. The authors should could consider cross-referencing the cases with meteorological data (e.g., radar, reanalysis).

**Reply:**

We changed the credibility categories tornado report on Table 1.

Table 1. Credibility and category tornado reports.

| Category | Criteria |
|---|---|
| Confirmed | 1. A photograph or video of a tornado
2. Damage survey of tornado |
| Probable | 1. Eyewitness who reported seeing rotation wind and/or column air
2. Eyewitness report of typical tornado damage
3. A photograph of a typical tornado damage |
| Possible | 1. Eyewitness who reported hearing thunder
2. There was hail
3. Cause of the damage is not confirmed by the observations of an eyewitness |

Regarding the cross-referencing of cases with meteorological data, such as radar and reanalysis, it is challenging to provide. The radar observation in Indonesia is limited, and some radars have areas blocked by topography. Thus, it is pretty challenging to provide cross-referencing for tornado climatology. The reanalysis data have a large spatial resolution that might not capture the characteristics of tornadoes. Furthermore, the time resolution of reanalysis with a 1-hour interval might not capture tornadoes with lifetimes on the minute scale.

3. Caution when interpreting a climate change signal in the occurrence of tornadoes. The spatial distribution shows a strong bias towards Java and very few reports from Maluku–Papua, which reflects both population and possibly data availability. This should be emphasised as a limitations. For example, in the Discussion section the

authors could add that provinces with few reports may simply lack observers or press coverage. One further suggestions is to have a separate Discussion section.

**Reply:**

We add the subsection 4.1 Limitation in section 4 Discussion.

"*The tornado cases were collected from four primary sources: (i) old newspaper archives from the website https://delpher.nl/, (ii) the National Library and Press Museum, (iii) the BMKG RI extreme weather database, and (iv) Social media. We realized that these sources can lead to a bias in the spatiotemporal distribution of tornado cases in Indonesia. Not all the old newspapers, both from the website and the National Library and Press Museum, were available for analysis. The description of the event was also limited, which could result in errors regarding location and time. The coverage of the old newspaper could result in bias in the spatial distribution of tornado cases, as we speculate in Section 3.2 that the old newspapers were primarily found in the big cities, such as Padang and Medan in North Sumatra, and Jakarta, Surabaya, and Bandung in Java. Reports from old newspapers may also have been influenced by editorial choices or public interest. The BMKG RI extreme weather database and social media could also result in bias, especially in spatial events, because it depends on the observer or people who reported the tornado. The population density in Indonesia is not distributed spatially (see Supplementary Files, Figure 1). For example, the Maluku-Papua and the central Kalimantan have population density around 0-50 person/km$^2$ compared to other regions, such as Java, that have a population density of over 500 person/km$^2$.*"

4.  The manuscript analyses two recent tornadoes (Rancaekek 2024 and Bogor 2018) using the EF-Scale and JEF-Scale. This shows that the same damage translates to a higher rating on the JEF-Scale (EF2 vs JEF3 for Rancaekek). The authors argue that differences in building practices cause this discrepancy, and suggest developing an Indonesian scale. However, two case studies are not enough to fully justify a new scale and the general conclusion.

**Reply:**

We acknowledge that two case studies are insufficient to justify the development of an Indonesian Tornado Scale. There is no damage assessment available after the tornado occurred, so there are no pictures or videos to show the damage and make a rating.

Moreover, the availability of photos or videos documenting tornado impacts is limited, and media coverage—particularly from internet news sources—is often sparse or lacks sufficient detail. However, this study provides a foundation for future efforts that Indonesia needs to establish standardized procedures for tornado assessment.

5. The Discussion mentions that most tornadoes occurred in the Nov–Mar season, aligning with the Austral summer monsoon. It also notes that effects of the MJO and ENSO are worth investigating. This is an important point as large-scale atmospheric modes strongly modulate convection over Indonesia. However, the manuscript does not analyse any of these factors quantitatively. A minimum improvement would be to cite previous studies that link convection in Indonesian to ENSO/MJO.

**Reply:**

We add the subsection 4.2 Influence of Convective Precursors in section 4 Discussion.

"*Previous studies have examined the influence of convective precursors on tornadic storms, such as the Madden-Julian Oscillation (MJO) (Barrett and Gensini, 2013; Tippett, 2018; Veloso-Aguila et al., 2024) and the El Niño–Southern Oscillation (ENSO) (Allen et al., 2015). Maritime Continent (MC) convective activity, especially Indonesia, is influenced by various phenomena, such as MJO, ENSO, Indian Ocean Dipole (IOD), monsoon, and cold surge (Yoneyama and Zhang, 2020). Muhammad et al. (2021) showed that during phases 2 through 4 of active MJO, convection develops in the MC and can increase the extreme precipitation probability. Kurniadi et al. (2021) found that ENSO and IOD can affect extreme precipitation in Indonesia. The impact of ENSO on extreme precipitation is more pronounced between June and November, but less significant from December to May. In contrast, the IOD has a significant effect on extreme precipitation only during the June to November period. Chang et al. (2005) showed that monsoon can influence the increasing and decreasing precipitation in MC. Cold surges also influence the precipitation in MC (Hattori et al., 2011). These phenomena modulate convective activity, which can subsequently lead to extreme precipitation. However, there are limited studies that specifically examine how synoptic-scale conditions modulate tornado activity in tropical regions. Therefore, further investigation is needed to assess the influence of convective precursors on tornado occurrence in Indonesia*"

**Technical Corrections**

- line 26: the phrase "a tornado events" should be "tornado events."
- lines 35-36: the term "basic climatology characteristics" should be "basic climatological characteristics."
- line 37: replace "catalyst future tornado studies" with "catalyze future tornado studies."
- line 47: "occuring" should be "occurring."

**Reply:**

Thank you for corrections. We already changed the words in the manuscript.

---

## Author Comment (AC2)

**Response to Reviewer #2**

We sincerely appreciate the time and effort you devoted to reviewing our work. Your insightful comments and constructive suggestions have been invaluable in enhancing the quality of our study. We have incorporated the comments and recommended suggestions. Please refer to the section below, marked in blue, for a detailed response to the reviewers' comments and concerns, with page numbers corresponding to the revised manuscript.

**General Comment:**

Overall: I'm always pleased to see additional information on tornadoes around the world. In the last 30 years, awareness has grown as to how widespread their occurrence is.

**Reply:** We thank to reviewer for reviewing and providing comments and suggestions. We provide point-by-point responses below.

**Specific Comments**

1. In the list of recently developed tornado databases, there is an additional one that should be included covering much of the former Soviet Union. Chernokulsky et al. (2020). Chernokulsky, A., and Coauthors, 2020: Tornadoes in Northern Eurasia: From the Middle Age to the Information Era. Mon. Wea. Rev., 148, 3081–3110, https://doi.org/10.1175/MWR-D-19-0251.1

**Reply:**

Thank you for the reference. We added in Line 19.

2. The inclusion of some metadata about the quality of the report is critical. It dates back to the origin of the European Severe Weather Database. Any additional information, particularly about the quality of witnesses, would be helpful. The difference between the opinions of, say, the study's authors upon seeing a tornado live compared to a child would be useful to know about.

**Reply:**

We changed the sentence:

"An eyewitness who reported observing rotating winds and/or a column of air was included in the 'probable' category, while an eyewitness who reported hearing thunder and observing hail was placed in the 'possible' category."

Into

"*An eyewitness testimony that reported observing rotating winds and/or a column of air directly was included in the 'probable' category, while an eyewitness testimony that reported hearing thunder and observing hail directly was placed in the 'possible' category.*"

The terms *testimony* and *directly* indicate that the eyewitness personally experienced the tornado event. In most reports, the eyewitness was an individual directly affected by the tornado, such as a resident of the impacted area during the event.

3. The discussion that changes in frequency are likely due to non-meteorological factors is useful to include. Given the apparent relatively rare occurrence of tornadoes in Indonesia, it is unlikely that the attribution to meteorological changes will ever occur. Even in the United States, with more than 1000 tornadoes per year, it is difficult to pull out the non-meteorological effects.

**Reply:**

We acknowledge that changes in frequency of tornado are likely due to non-meteorological factors. Thus, we added the limitation of this study in the discussion.

We add the subsection 4.1 Limitation in section 4 Discussion.

4. I don't see much of a need to develop a new damage scale. The sample size superimposed on the relatively rare nature of the events makes it hard to have much confidence in meaningful information. If it is possible to find relatively similar construction practices in other countries, it might be possible to get a significant sample, but I doubt it. Even in the United States, there are serious problems with ratings of tornadoes, e.g., Lyza et al. (2025) Lyza, A. W., H. E. Brooks, and M. J. Krocak, 2025: Where Have the EF5s Gone? A Closer Look at the "Drought" of the Most Violent Tornadoes in the United States. Bull. Amer. Meteor. Soc., https://doi.org/10.1175/BAMS-D-24-0066.1, in press.

**Reply:**

We acknowledge the challenges of developing a new damage scale in Indonesia due to the rarity of tornadoes and the small sample size. Even in the United States, rating uncertainties persist (e.g., Lyza et al., 2025). However, an Indonesia-specific scale is crucial because local construction practices, building materials, and structural vulnerabilities differ from those in countries where existing scales were developed. Despite the limited data available, a tailored scale would enhance rating accuracy, support disaster preparedness, and establish a consistent framework for future climatological and comparative studies. Therefore we include in the discussion of this study for giving future suggestions for Indonesian government.